# Antigen Targeting of Porcine Skin DEC205^+^ Dendritic Cells

**DOI:** 10.3390/vaccines10050684

**Published:** 2022-04-26

**Authors:** Edgar Alonso Melgoza-González, Mónica Reséndiz-Sandoval, Diana Hinojosa-Trujillo, Sofía Hernández-Valenzuela, Melissa García-Vega, Verónica Mata-Haro, Araceli Tepale-Segura, Laura C. Bonifaz, Armando Perez-Torres, Jesús Hernández

**Affiliations:** 1Laboratorio de Microbiología e Inmunología, Centro de Investigación en Alimentación y Desarrollo AC, Hermosillo 83304, Mexico; edgaralonso.mglez@gmail.com (E.A.M.-G.); mresendiz@ciad.mx (M.R.-S.); diana.hinojosat@gmail.com (D.H.-T.); sofiahdzval@gmail.com (S.H.-V.); melsgv23@gmail.com (M.G.-V.); vmata@ciad.mx (V.M.-H.); 2Unidad de Investigación Médica en Inmunoquímica, Hospital de Especialidades, Centro Médico Nacional Siglo XXI, Instituto Mexicano del Seguro Social, Ciudad de Mexico 06720, Mexico; tepale.araceli@gmail.com (A.T.-S.); labonifaz@yahoo.com (L.C.B.); 3Departamento de Inmunologia, Escuela Nacional de Ciencias Biológicas del Instituto Politécnico Nacional, Ciudad de México 07738, Mexico; 4Laboratorio de Filogenia del Sistema Inmune y Mucosas, Departamento de Biología Celular y Tisular, Universidad Nacional Autónoma de México, Ciudad de Mexico 04510, Mexico; armandop@unam.mx

**Keywords:** antigen targeting, DEC205, porcine, dendritic cells, cDC1, cDC2

## Abstract

Dendritic cell (DC) targeting by DEC205^+^ cells effectively promotes the internalization of antigens that may trigger a specific immune response. In this study, we evaluated the ability of a recombinant antibody, anti-DEC205 (rAb ZH9F7), to trigger cellular endocytosis in subpopulations of DCs and targeted cells after intradermal injection and subsequent migration toward lymph nodes. Furthermore, the cellular immune response was evaluated in pigs after intradermal application of the antigenized rAb ZH9F7 combined with porcine circovirus type 2 cap antigen (rAb ZH9F7-Cap). We demonstrated that rAb ZH9F7 recognized conventional type 1 and 2 DCs from the blood and skin and monocytes. It promoted receptor-mediated endocytosis and migration of cDCs and moDCs toward regional lymph nodes. Intradermal application of rAb ZH9F7-Cap induced a higher frequency of IFN-γ-secreting CD4^+^CD8^+^ T lymphocytes and antibodies against Cap protein than that in the control group. In conclusion, the rAb ZH9F7-Cap system promoted the target of skin cDC1 and cDC2, provoking migration to the regional lymph nodes and inducing a Th1 response, as evidenced by the proliferation of double-positive CD4^+^CD8^+^ T cells, which correlates with an enhanced ability to target the cDC1 subset both in vitro and in vivo.

## 1. Introduction

Prophylactic and therapeutic antibodies have been explored for over a decade, and their use in antigen targeting is a popular strategy. This method delivers antigens to antigen-presenting cells through antibodies that recognize surface receptors to promote capture, processing, and presentation to naïve T lymphocytes [1]. The intrinsic ability of dendritic cells (DCs) to perform these processes makes them an attractive vaccine development target based on antigen targeting. Several authors have evaluated the effect of antigen targeting on swine models through different routes of administration such as intravenous, intramuscular, subcutaneous, and intradermal administration [2,3,4].

Currently, the presence of monocyte-derived DCs (moDCs), plasmacytoid DCs (pDCs), and conventional type 1 and 2 DCs (cDC1 and cDC2) has been identified in several species, including pigs [5,6]. These DC subpopulations are phenotypically different and exhibit different functions shared between species [7]. Porcine moDCs (in vitro derived from blood monocytes cultured with IL-4 and GM-CSF) can stimulate CD4^+^ T lymphocytes toward a Th1 response, leading to the production of proinflammatory cytokines [8]. Both cDC1 and cDC2 can take up soluble antigens in porcine blood, although the cDC2 population has enhanced endocytic activity skewed toward a Th2 response. In contrast, the porcine cDC1 subset shows a stronger ability to cross-present, thus stimulating CD8^+^ T lymphocytes for IFN-γ production [9].

To improve DC cross-presentation, researchers have studied the C-type lectin family member receptor DEC205 with different strategies. This receptor has shown positive effects in stimulating antigen-specific T lymphocytes, achieving antitumoral and antiviral immune responses [3,10,11,12]. For example, Subramaniam et al., (2017) evaluated antigens of porcine respiratory and reproductive syndrome virus (PRRSV) targeting DEC205+ DCs in pigs. They found that this strategy promoted the production of IFN-γ and IL-4 by CD4+ T lymphocytes. Nonetheless, the authors observed poor immunogenicity in their DC-targeting vaccine, probably due to intramuscular application, since the presence of DC populations is scarce in this region [2].

In porcine skin, different DC subsets have been previously characterized using several markers: CD172a^+^ CD163^−^ CADM^+^, CD172a^−/low^ CD163^−^ CADM^+^, CD172a^+^ CD163^low^ CADM^−/low^ and CD172a^+^ CD163^high^ CADM^−^ phenotypically compatible with Langerhans cells (LCs), cDC1, cDC2 and moDCs, respectively [13]. Moreover, it has been demonstrated that porcine LCs, cDC1, and cDC2 exhibit mobile capabilities in a steady and inflammatory state, whereas moDCs migrate only under lymph inflammation [14]. Furthermore, LCs and dermal DCs are susceptible to in situ labeling in both mice and humans by the mAb-anti DEC205 after intradermal application [15]. In summary, skin DCs show promise for applying vaccines based on antigen targeting to DCs.

Previously, our research group developed a recombinant mouse × pig chimeric antibody against porcine DEC205 receptor (rAb ZH9F7), and antigens from PRRSV were targeted by intradermal application [16,17]. The obtained results showed that the system was immunogenic with limited protective immunity. Therefore, several questions remain regarding how the targeting of DCs DEC205^+^ can induce protective immunity and how it could be optimized for use against other pathogens such as porcine circovirus type 2 (PCV2).

PCV2 belongs to the *Circoviridae* family and is an icosahedral nonenveloped virus with a circular single-stranded DNA genome of fewer than 2000 nucleotides in length; therefore, it is considered to be the smallest member of the family [18]. Among the 11 potential open reading frames (ORFs) on the PCV2 genome is ORF2, which encodes the capsid protein (Cap), the major structural component and the main target of humoral and cellular immune responses [19,20]. These characteristics make PCV2 Cap a suitable model for evaluating the targeting of antigens to DEC205^+^ DCs.

In this study, we characterized antigen targeting in a swine model and evaluated the ability of the rAb ZH9F7 to be internalized by DEC205^+^ DCs in peripheral blood. Moreover, we intradermally applied the rAb ZH9F7 recombinant antibody and evaluated the targeting of dermal cDC1 and cDC2 and the presence of targeted DCs in regional lymph nodes. Finally, the antigenized rAb ZH9F7-Cap promoted antibody production against PCV2a Cap and the differentiation of double-positive CD4^+^CD8^+^ cells that respond to stimulation with rAb ZH9F7-Cap through IFN-**γ** production characterized by its strong antiviral capabilities.

## 2. Materials and Methods

### 2.1. Animals

Nineteen pigs included in this study were obtained from PRRSV and influenza virus-free farms without a PCV2 history. Conventional piglets (female and male) arrived at 3–4 weeks of age in healthy condition. They were housed in the experimental facility of Centro de Investigación en Alimentación y Desarrollo A. C. (CIAD), with water and food access ad libitum. Before inoculation, the pigs had very low anti-PCV2 antibodies measured by ELISA (absorbance < 0.200). The protocols applied were approved by the CIAD ethics committee (CE/020/2014). At the end of the trials, the pigs were euthanized by the recommendations of the Mexican Official policy NOM-033-ZOO-1995.

### 2.2. Design, Expression, and Characterization of rAb ZH9F7-Cap

The chimeric mouse × pig anti-DEC205 antibody design was performed as previously reported by Bustamante-Córdova et al. [17]. The synthesis of genes encoding SwHCZH9F7-Cap and SwLCZH9F7 was performed by GenScript (Piscataway, NJ, USA). All genes were cloned into the pcDNA3.1(-) expression vector: pcDNA3.1(-)/SwHCZH9F7-Cap and pcDNA3.1(-)/SwLCZH9F7. An isotype control antibody was designed to replace the VH and VL sequences from pcDNA3.1(-)/SwHCZH9F7-Cap and pcDNA3.1(-)/SwLCZH9F7 with VH and VL sequences from an irrelevant monoclonal antibody (mAb 7DEU). The obtained plasmids were pcDNA3.1(-)/SwHC7DEU-Cap and pcDNA3.1(-)/SwLC7DEU.

To produce the rAb ZH9F7 antibody and rAb 7DEU isotype control antibody combined with PCV2a Cap (hereafter referred to as rAb ZH9F7-Cap or rAb 7DEU-Cap, respectively), the Expi293 Expression System (Thermo Fisher Scientific, Waltham, MA, USA) was chosen, and the expression was performed according to the manufacturer’s instructions. Briefly, 7.5 × 10^7^ Expi293 cells were seeded in 25.5 mL of Expi293 Expression Medium. The transfection complex was prepared using 30 µg of pcDNA3.1(-)/SwHCZH9F7-Cap mixed with pcDNA3.1(-)/LCZH9F7 (ratio 1:2) and 81 µL of Expifectamine, and both were diluted in Opti-MEM medium (Thermo Fisher Scientific, Waltham, MA, USA). The exact ratio was used for transfection with the plasmids pcDNA3.1(-)/SwHC7DEU-Cap and pcDNA3.1(-)/SwLC7DEU. Transfected cells were incubated at 37 °C, 8% CO_2_, and 125 rpm. Twenty hours post-transfection, enhancers 1 and 2 were added to the culture flasks. Daily sampling and determination of cell viability by trypan blue exclusion staining (Sigma-Aldrich, St. Louis, MO, USA) was performed. Harvesting was performed four days post-transfection, and the culture supernatants were clarified by centrifugation at 493× *g* for 10 min. According to the manufacturer’s instructions, the clarified supernatants were loaded into HiTrap Protein G HP (GE Healthcare Life Sciences, Chicago, IL, USA). Fractions of 1 mL were collected and analyzed by native 10% PAGE followed by Coomassie blue staining (Bio-Rad, Hercules, CA, USA). Characterization of purified rAb ZH9F7-Cap and rAb 7DEU-Cap was performed by western blot using HRP-conjugated goat anti-porcine IgG H + L (Southern Biotech, Birmingham, AL, USA) 1:5000 for 1 h at 37 °C, followed by the addition of 3,3′,5,5′ tetramethylbenzidine (TMB) membrane substrate system (Sigma, St. Louis, MO, USA). Additionally, a recombinant rAb ZH9F7 coupled to an enhanced green fluorescent protein (rAb ZH9F7-eGFP) was produced to evaluate targeting by confocal microscopy.

### 2.3. Peripheral Blood Mononuclear Cell (PBMC) Isolation

All in vitro experiments of DE205^+^ cell recognition, internalization, and stimulation were performed using PBMCs. Once EDTA-treated blood samples were taken, PBMCs were isolated by a Ficoll-Paque plus density gradient. Briefly, 4 mL of EDTA blood samples were carefully added into a 15 mL tube with 8 mL of Ficoll reagent using a blood-Ficoll-Paque plus ratio (GE Healthcare, Plainfield, NJ, USA) of 1:2. Then, the tubes were centrifuged at 493× *g* for 30 min without a break. After centrifugation, PBMCs were carefully taken and washed twice with RPMI 1640 (Sigma-Aldrich, St. Louis, MO, USA) supplemented with an antibiotic-antimycotic cocktail (penicillin–streptomycin Sigma-Aldrich, St. Louis, MO, USA: 100 units/mL and 100 μg/mL, respectively; amphotericin B solution Sigma-Aldrich, St. Louis, MO, USA: 0.5 μg/mL; gentamicin Invitrogen, Waltham, MA, USA: 50 μg/mL), 10% FBS, and two mM EDTA). Finally, the cells were resuspended in complete RPMI 1640 (Sigma-Aldrich, St. Louis, MO, USA) until assays were performed.

### 2.4. Internalization Assay with rAb ZH9F7 and Functionality Evaluation of rAb ZH9F7-Cap and rAb7DEU-Cap

To evaluate DEC205 receptor-mediated endocytosis in the subpopulations of blood DCs, we performed an internalization assay of the rAb ZH9F7 by flow cytometry. First, 1 mg of rAb ZH9F7-Cap was biotinylated according to the manufacturer’s instructions of the EZ-Link™ Sulfo-NHS-LC-Biotinylation Kit (Thermo Fisher Scientific, Waltham, MA, USA). Then, a sample of porcine blood was taken by venipuncture, and PBMCs were isolated through density gradient Ficoll-Paque. After Fc receptor blocking (with phosphate-buffered saline 1X pH 7.4, 0.1% bovine serum albumin, and 10% porcine serum), 1 × 10^6^ cells were incubated at room temperature for 15 min with 0.1 μg of biotinylated rAb ZH9F7. Then, the cells were washed, and mouse IgG2b anti-CD14 (Bio-Rad, Watford, UK), chicken IgY anti-CADM1 (MBL International, Woburn, MA, USA), and mouse IgG1 anti-CD172 (Bio-Rad, Watford, UK) antibodies were added and incubated for 15 min at 4 °C. Finally, FITC-conjugated goat anti-mouse IgG2b (Southern Biotech, Birmingham, AL, USA), Alexa Fluor 647-conjugated goat anti-chicken IgY (Invitrogen, Waltham, MA, USA), BV421-conjugated goat anti-mouse IgG1 (BioLegend, San Diego, CA, USA) and streptavidin PE-conjugated (Southern Biotech, Birmingham, AL, USA) were added and incubated for 15 min at 4 °C. After incubation with the biotinylated rAb ZH9F7, PBMCs were incubated for 30 and 60 min at 37 °C to internalize the rAb ZH9F7-DEC205 complex, and the assay was performed as described above. As a control, cells were kept on ice for 30 and 60 min to allow basal internalization. In addition, the same staining panel evaluated the recognition capacity of rAb ZH9F7-Cap and rAb 7DEU-Cap toward DEC205^+^ cells in porcine peripheral blood without performing internalization kinetics. Following the manufacturer’s instructions, both antibodies were RPE-conjugated using a LYNX rapid RPE antibody conjugation kit (Bio-Rad, Hercules, CA, USA). All experiments were performed in a FACS Aria III flow cytometer (BD Biosciences, NJ, USA) and analyzed using FACS Diva Software version 8.0.1 (BD Biosciences, NJ, USA). At least 1 × 10^6^ cells were acquired in each experiment. The gating strategies for each experiment are described in Appendix A. All antibodies and dilutions used are listed in Appendix A.

### 2.5. In Vivo DEC205^+^ DC Targeting by rAb ZH9F7-Cap

To evaluate the ability of rAb ZH9F7-Cap to recognize and target DCs in peripheral tissues, 60 μg of RPE-conjugated rAb ZH9F7-Cap and 100 μg of poly (I:C) adjuvant (InvivoGen, San Diego, CA, USA) were mixed and diluted in sterile phosphate-buffered saline (PBS, pH = 7.4) to a final volume of 600 μL. The RPE-conjugated rAb 7DEU-Cap was used as an isotype control mixed with 100 μg of poly (I:C) in the same final volume. The formulations were injected intradermally using a hypodermic needle in the right inguinal area of three healthy pigs (per group). As negative controls, three healthy pigs were injected with PBS (phosphate-buffered saline, pH 7.4) and poly (I:C) adjuvant. After 24 h, pigs were stunned by electric shock and later euthanized by exsanguination under the Mexican official policy Nom-033-ZOO-1995. Skin biopsies were obtained from the injection site (right inguinal area) and both superficial inguinal lymph nodes (right and left).

### 2.6. Analysis of Targeted DEC205^+^ Cells in the Skin and Regional Lymph Nodes

Skin biopsies were cultured in RMPI-1640 (Sigma-Aldrich, St. Louis, MO, USA) supplemented with 10% FBS, collagenase D (1 mg/mL, Worthington Biochemical Corporation, USA), dispase (0.5 mg/mL, Invitrogen, Waltham, MA, USA) and antimycotic-antibiotic cocktail and incubated at 37 °C and 8% CO_2_ for 22 h. Skin migratory cells in the culture supernatant were harvested and filtered through 100 μm cell strainers. After cell counting, 5 × 10^5^ migratory cells were blocked (PBS + 10% pig serum) and stained with mouse IgG1 anti-CD163 (Bio-Rad, Watford, UK), chicken IgY anti-CADM1 (MBL International, Woburn, MA, USA), and CTLA-4 (Ancell, Bayport, MN, USA) for 15 min at room temperature. After the cells were washed, BV421-conjugated goat anti-mouse IgG1 (BioLegend, San Diego, CA, USA), Alexa Fluor 647-conjugated goat anti-chicken IgY (Invitrogen, Waltham, MA, USA), FITC-conjugated mouse anti-CD172a (Bio-Rad, Watford, UK) and PE/Cy7 streptavidin (BioLegend, San Diego, CA, USA) were added and incubated for 15 min at room temperature. For the experiment mentioned above, skin biopsies of pigs injected with PBS and poly (I:C) adjuvant were used as negative controls for DC targeting.

Targeting of DCs in regional lymph nodes was evaluated. After pigs were euthanized, superficial inguinal lymph nodes were removed and macerated through 100-μm pore-nylon mesh filters. Cells were placed on RPMI 1640 (Sigma-Aldrich, St. Louis, MO, USA) supplemented with 10% FBS and antibiotic-antimycotic cocktail, and enrichment of DCs was performed by an OptiPrep (Sigma-Aldrich, Darmstadt, Germany) density gradient. DC enrichment was performed with the OptiPrep gradient protocol following modifications previously established by Puebla-Clark et al., 2019 [21]. Briefly, the manufacturer’s protocol on application sheet C20 was modified using RPMI 1640 medium (Sigma-Aldrich, St. Louis, MO, USA) to dilute 11.5% iodixanol, and the suspension solution was replaced by Hanks’ balanced salt solution supplemented with 5% fetal bovine serum and 2 mM EDTA.

After Fc receptors were blocked, 3 × 10^6^ cells were stained with IgG1 mouse anti-CD3 (Southern Biotech, Birmingham, AL, USA), IgG1 mouse anti-CD21 (Southern Biotech, Birmingham, AL, USA), and IgY chicken anti-CADM1 (MBL International, Woburn, MA, USA) at room temperature for 15 min. Then, BV421-conjugated goat anti-mouse IgG1 (BioLegend, San Diego, CA, USA) and Alexa Fluor 647-conjugated goat anti-chicken IgY (Invitrogen, Waltham, MA, USA) were added and incubated at room temperature for 15 min. Cells were washed after each incubation step, and Alexa Fluor 488-conjugated IgG1 mouse anti-CD163 (Bio-Rad, Watford, UK) was added in a final step. Labeled cells from the skin and lymph nodes were acquired in a FACS Aria III flow cytometer (BD Biosciences, NJ, USA) and analyzed using FACS Diva Software version 8.0.1 (BD Biosciences, NJ, USA).

### 2.7. Confocal Analysis of rAb ZH9F7-eGFP Internalization by Skin DCs

Pigs were injected with 60 μg of rAb ZH9F7-eGFP + 100 μg of poly (I:C) intradermally or sterile PBS + 100 μg of poly (I:C). After 6 h, skin biopsies were taken with an 8 mm disposable skin punch. PBS. Skin biopsies were fixed with formaldehyde (Sigma-Aldrich, St. Louis, MO, USA) and then embedded in paraffin. Sections of 5 µm from control (PBS) or targeted (rAb ZH9F7-eGFP) skin were placed on charged glass slides (Superfrost Plus Green). Slides were incubated on a stove (70 °C) for 40 min to remove excess paraffin. Tissues were rehydrated with a xylene/ethanol solution of solvents (Sigma-Aldrich, St. Louis, MO, USA). Antigen retrieval was performed using citrate buffer pH 6.0 (10 mM sodium citrate) at 120 °C for 20 min. Skin samples were permeabilized for 2 h with a perm solution containing 10 mg/mL bovine serum albumin, 5% horse serum (Biowest, Riverside, MO, USA), 0.02% sodium azide, and 0.3% Triton-100 (Sigma-Aldrich, St. Louis, MO, USA). After permeabilization, samples were incubated with primary antibodies: anti-CADM1 (MBL International, Woburn, MA, USA) for 18 h. Then, secondary Alexa Fluor 647-conjugated goat anti-chicken IgY (Invitrogen, Waltham, MA, USA) and mouse IgG1 anti CD172a (Bio-Rad, Watford, UK) RPE-conjugated (Invitrogen, Waltham, MA, USA) were added and incubated for 2 h. Nuclei were stained with Hoechst (Invitrogen, Waltham, MA, USA) for 10 min. Sections were mounted with Vectashield (Vector Laboratories, Burlingame, CA, USA) and stored at 4 °C. Micrographs were obtained on a Nikon Ti Eclipse inverted confocal microscope (Nikon Corporation, Tokyo, Japan) using NIS Elements v.4.50. Imaging was performed using a 20× (dry, NA 0.8) objective lens. Evaluation was performed at 3.4×, and Digital Zoom was used when specified. Images were analyzed using FIJI ImageJ software (ImageJ software, National Institutes of Health; http://rsbweb.nih.gov/ij/, accessed on 22 February 2022).

### 2.8. Immunization

To determine whether targeting PCV2 Cap to DCs through rAb ZH9F7 can prime the cellular immune response, 8-week-old pigs (*n* = 3) were intradermally vaccinated with rAb ZH9F7-Cap (doses equivalent to 100 μg of Cap mixed with 100 μg of poly (I:C)) in the right inguinal area. The control group (*n* = 2) consisted of pigs of the same age who did not receive the rAb ZH9F7-Cap vaccine. Three weeks after the first immunization, a boost was applied at the same dose and route of administration. One week after the booster, 6 mL of peripheral blood sample was taken into EDTA tubes to evaluate the response of IFN-γ-secreting cells in PBMCs. Moreover, serum samples were obtained at 0 and one week after boosting to analyze total IgG anti-Cap antibodies.

### 2.9. Antibody Response

The humoral immunity induced by immunization with rAb ZH9F7-Cap was evaluated by indirect ELISA. Briefly, a high-adherence 96-well ELISA plate was coated with recombinant PCV2a Cap antigen (2 μg/mL) diluted in carbonate buffer (100 mM, pH 9.6) at 4 °C overnight. After incubation, the plate was washed once with PBS 1X. Then, the plate was blocked for 1 h at room temperature with 2% bovine serum albumin in PBST (PBS 1X pH with 0.1% Tween-20). Swine serum samples were diluted at 1:100 in PBS with 25% goat sera, and 50 μL of the diluted samples were added to the ELISA plate in triplicate. The samples were incubated for 30 min at room temperature with gentle agitation. After incubation was complete, the samples were washed five times with PBST buffer. HRP-conjugated goat anti-porcine IgG H + L (Southern Biotech, Birmingham, AL, USA) was diluted in PBST (1:5000), and 50 μL was added to each well of an ELISA plate and incubated for 30 min at room temperature with constant agitation. Five washes with PBST were performed, and 50 μL of TMB 1-Component HRP Microwell Substrate (ImmunoChemistry Technologies, Davis, CA, USA) was added to each well. After 5 min of incubation in darkness at room temperature, 50 μL of H_2_S0_4_ (1 M) was added to stop the colorimetric reaction. Finally, the optical density was measured using a microplate reader at 450 nm.

### 2.10. Stimulation of PBMCs for IFN-γ Production

To evaluate the production of the IFN-γ response in vitro, PBMCs were isolated by Ficoll density gradient as previously described. For IFN-γ stimulation, 1 × 10^6^ cells were seeded in 48-well plates with 500 μL of RPMI 1640 (Sigma-Aldrich, St. Louis, MO, USA). complete with 10% FBS, 2-mercaptoethanol (Sigma-Aldrich, St. Louis, MO, USA), and antimicrobial cocktail with phytohemagglutinin (PHA, Sigma-Aldrich, St. Louis, MO, USA) at 10 μg/mL, rAb ZH9F7-Cap (36 μg/mL equivalent to 10 μg/mL Cap), rAb ZH9F7 as a control (26 μg/mL) and medium only. Then, the cells were incubated at 37 °C and 5% CO_2_ for 8 h. After incubation, a protein transport inhibitor cocktail (500X) (Invitrogen, Waltham, MA, USA) was added to a final dilution of 1X. Cells were incubated for an additional 16 h before harvesting. Cells were blocked with 10% porcine serum in FACS buffer. For extracellular staining, IgG2a mouse anti-pig CD8α (Bio-Rad, Watford, UK) and IgG2b mouse anti-pig CD4α (Bio-Rad, Watford, UK) were added and incubated for 15 min at room temperature. Following washing, FITC-conjugated goat anti-mouse IgG2a (Southern Biotech, Birmingham, AL, USA) and Alexa Fluor 647-conjugated goat anti-mouse IgG2b (Invitrogen, Waltham, MA, USA) were added and incubated for an additional 15 min. For intracellular staining, cells were fixed and permeabilized with a Leucoperm kit (Bio-Rad, Hercules, CA, USA) following the manufacturer’s instructions, and intracellular IFN-γ was labeled by the addition of PE-conjugated mouse anti-pig IFN-γ (BD Biosciences Pharmingen, San Diego, CA, USA).

Finally, the cells were acquired and analyzed in a FACS Aria III cytometer (BD) and analyzed using FACS Diva Software version 8.0.1 (BD). The percentage of IFN-γ measured in unstimulated cells was subtracted to eliminate basal IFN-γ production for all experiments. Although there are other more common methods such as ELISPOT or ELISA to evaluate the frequency of IFN-γ secreting cells and amount of IFN-γ, we decided to conduct this analysis through flow cytometry. The main advantage of using this method was that we could analyze the phenotype of the cells producing IFN-γ.

### 2.11. Statistical Analysis

All analyses and graphics were performed in GraphPad Prism version 9.0. Normality was determined using the Shapiro–Wilk test. One-way (or two-way, as indicated) ANOVA was performed to evaluate differences between the treatment means, followed by multiple mean comparisons by the Tukey–Kramer test. Statistically significant differences were considered with a *p*-value < 0.05.

## 3. Results

### 3.1. Expression of rAb ZH9F7-Cap and rAb 7DEU-Cap, Purification, and Characterization

After Expi293 cells were transfected, viability monitoring was performed daily, and the cells were harvested on day four post-transfection when the percentage of viable cells was approximately 60%. Figure 1a shows a schematic illustration of the rAb ZH9F7-Cap and isotype control rAb 7DEU-Cap. Analysis of both purified rAb ZH9F7-Cap and rAb 7DEU-Cap is shown in Figure 1b. As expected, lane 2 shows the estimated bands of rAb ZH9F7 without Cap at approximately 25 and 50 kDa for the heavy and light chains, respectively. Lanes 3 and 4 show two bands with predicted molecular weights of 75 and 25 kDa corresponding to the heavy and light chains of rAb ZH9F7-Cap and rAb 7DEU-Cap. Moreover, to confirm that rAb ZH9F7-Cap and rAb 7DEU-Cap were indeed mouse × pig chimeric antibodies, we performed a western blot to detect porcine IgG (H + L). In Figure 1c, lane 3 corresponds to the purified rAb ZH9F7-Cap, lane 4 corresponds to rAb 7DEU-Cap, where two predominant bands of 25 and 75 kDa were visible, and lane 2 shows rAb ZH9F7 without Cap antigen.

### 3.2. rAb ZH9F7-Cap Recognizes Blood DEC205^+^ Cells, including cDC1, cDC2, and Monocytes, and Triggers Receptor-Mediated Endocytosis

The population of interest was gated to evaluate the ability of the three rAbs produced to recognize DEC205+ cells, as shown in Figure 2a. The gating strategy and fluorescence minus one (FMO) control are shown in Appendix A. Consistent with a previous report by Parra-Sanchez et al. [21], rAb ZH9F7 identified cDC1 (CD14^−^CADM1^+^CD172a^low^DEC205^+^), cDC2 (CD14^−^CADM1^+^CD172a^high^DEC205^+^) and monocytes (CD14^+^DEC205^+^) on porcine PBMCs (Figure 2a). Figure 2b shows the expression of DEC205 on cDC1, cDC2, and monocytes detected by rAb ZH9F7, rAb ZH9F7-Cap, and isotype control rAb 7DEU-Cap.

Both rAb ZH9F7 and rAb ZH9F7-Cap showed a high percentage of DEC205 recognition in the cDC1, cDC2 and monocyte populations (above 84%, 66% and 59%, respectively) compared to the isotype control (*p* < 0.05) (Figure 2c). These findings confirm that rAb ZH9F7-Cap can recognize DEC205^+^ cDC1, cDC2, and monocytes similarly to its counterpart without the Cap antigen.

To demonstrate that rAb ZH9F7 promotes DEC205 internalization in porcine cells, cells were labeled with biotinylated-rAb ZH9F7 and incubated at 37 °C to enhance receptor-mediated endocytosis. The obtained results showed a reduced fluorescence intensity on cDC1, cDC2, and monocytes in a time-dependent manner compared with control cells (Figure 3a).

After graphing the percentage of reduction of mean fluorescence intensity (MFI), a decrease of the initial fluorescence of up to 36% for cDC1, 46% for cDC2, and 31% for monocytes was observed (Figure 3b). The fluorescence reduction is due to the internalization of DEC205 on the cell surface, which was triggered by the binding of rAb ZH9F7 to the DEC205 receptor at 37 °C. There were significant differences in the MFI reduction of each population of DEC205^+^ cells after 60 min of incubation at 37 °C compared with those after 60 min of incubation at 4 °C. However, there were no differences in receptor-mediated endocytosis among cDC1, cDC2, and monocytes after 60 min of incubation.

### 3.3. rAb ZH9F7-Cap Recognizes Different Skin DC Populations and Reaches Regional Lymph Nodes

Once we confirmed that rAb ZH9F7 specifically recognizes subsets of DCs (cDC1 and cDC2) and triggers a receptor-mediated endocytosis process in cDC1, cDC2, and monocytes from porcine blood, we evaluated the ability of rAb ZH9F7-Cap to target different populations of DCs in pig skin and thus confirmed its functionality in vivo. After rAb ZH9F7-Cap or rAb 7DEU-Cap (isotype control) was injected, skin biopsies were taken, and lymph node cells were harvested. Cells were analyzed for CD163 and CD172a expression, identifying possible moDCs and macrophages (moDCs/Macro) with the CD163^+^ CD172a^+^ phenotype. Then, the CD163^−^ CD172a^+/−^ population was analyzed for CADM1 and CD172a expression, identifying cDC1 (CD163^−^ CD172a^−^ CADM1^+^) and both cDC2 and possible LC populations (CD163^−^ CD172a^high^ CADM1^low/+^) (Figure 4a). The gating strategy and FMO controls are shown in Appendix A.

In Figure 4b, dot plots represent the results from 1 of 3 independent experiments. Migratory cells from the skin injection site of rAb ZH9F7-Cap showed phenotypes of cDC1, cDC2/LC, and moDCs/Macro positive for rAb ZH9F7-Cap labeling (12.87%, 9.1%, and 5.56% means, respectively). Migrating cDC1, cDC2/LC, and moDCs/Macro from skin biopsies of the control group (PBS) presented background fluorescence values of 4.53%, 4.66%, and 4.03%, respectively. Likewise, the percentages found for the isotype control rAb 7DEU group were 3.26%, 8.03%, and 6.66%, respectively. Statistically, differences between cDC1-targeted cells were found between rAb ZH9F7-Cap and both control groups (*p* < 0.05). The percentage of cDC1-targeted cells in the rAb ZH9F7-Cap groups was significantly higher than that in both control groups (*p* < 0.05). Additionally, cDC2-targeted cells from the rAb ZH9F7-Cap group were higher than those from the non-targeted group (*p* < 0.05) (Figure 4c). However, there were no significant differences in moDCs/Macro positive in either group (*p* > 0.05). Finally, the evaluation of activation marker expression CD80/86 on different skin-targeted populations was performed, albeit no statistically significant differences (*p* > 0.05) were found between groups (Appendix A).

Once we confirmed that rAb ZH9F7-Cap recognizes and targets skin cDC1 and cDC2 populations, we evaluated the ability of these cells to reach superficial inguinal lymph nodes, a site where antigen presentation to naïve T cells occurs. Figure 5a presents the strategy for selecting the population of interest. Lymph node cells had a high number of lymphocytes expressing the DEC205 receptor. We labeled CD3 and CD21 receptors to rule out T and B lymphocytes, respectively, to exclude them from the analysis. Subsequently, the CD3^−^CD21^−^ population was analyzed for CADM1 and CD163 expression. Then, CD3^−^CD21^−^CADM1^+^CD163^−^ cells were classified as potential DCs, and CADM1^+/−^CD163^+^ cells were classified as probable macrophages. The gating strategy and FMO control are shown in Appendix A.

Figure 5b shows histograms representative of one of three independent experiments. The targeting with rAb ZH9F7-Cap exhibits the presence of labeled DEC205^+^ DCs, macrophages, and lymphocytes in superficial regional inguinal lymph nodes, contrary to both control groups. Additionally, we observed that the presence of targeted DCs, macrophages, and lymphocytes was localized predominantly in the right superficial inguinal lymph node compared with the left side (data not shown). Figure 5c shows the average percentages of DCs, macrophages, and lymphocytes. The presence of targeted DCs from the rAb ZH9F7-Cap group (12.33%) was significantly higher than that from the control (2.73%) and isotype (5.83%) groups. However, there were no significant differences between targeted macrophages and lymphocytes in either group (*p* > 0.05). Thus, the intradermal application of rAb ZH9F7-Cap enables the identification of labeled dermal DEC205^+^ DCs that reach local lymph nodes.

### 3.4. Immunofluorescence from Skin Biopsies on the rAb ZH9F7-eGFP Targeting Group

The objective is to demonstrate that rAb anti-DEC205 not only recognizes DEC205^+^ cells but is also internalized after DEC205^+^ binding. When the micrographs were taken and analyzed, a large amount of cellular infiltrate was observed, which was expected due to the application of the inflammatory stimulus poly (I:C) (Figure 6a, lower panel) and positivity for the rAb ZH9F7-eGFP (green cells, green arrows). Subsequently, it was possible to find the presence of cDC1 (pink arrows) and cDC2 (green arrows), which were labeled in situ with the recombinant rAb ZH9F7 anti-DEC205 antibody conjugated with eGFP. Figure 6b shows two different fields wherein in vivo targeting of skin cDC1 and cDC2 was confirmed. These findings support the results on receptor-mediated endocytosis (from blood cDCs and monocytes), where the evidence is demonstrated by colocalization of fluorescent staining signals on skin cDC1- and cDC2-targeted cells.

### 3.5. Humoral Immune Response

The humoral immune response triggered by the inguinal intradermal application of rAb ZH9F7-Cap was evaluated four weeks post-injection (one week after boost). The immunization schedule is shown in Figure 7a. The bars in Figure 7b present the average optical density (O.D.) of IgG antibody titers against PCV2a Cap antigen. At week four, IgG antibodies were detected in the targeted group compared to the control group (*p* < 0.05).

### 3.6. Stimulation of PBMCs for IFN-γ Production

After the cells were harvested, stained, and acquired, CD4^+^CD8^−^, CD4^−^CD8^+^, and CD4^+^CD8^+^ populations producing IFN-γ were identified, as shown in Appendix A. Of note, the double-positive population represents a significant proportion of total T lymphocytes. Then, the production of IFN-γ in response to a selective T lymphocyte mitogen such as PHA was evaluated. Figure 7c represents the production of IFN-γ in the three identified T-cell populations, where double-positive CD4^+^CD8^+^ cells are the main IFN-γ producing cells, followed by CD4^−^CD8^+^ and CD4^+^CD8^−^ cells. The bar graph in Figure 7c shows that, regardless of the group (vaccinated with rAb ZH9F7-Cap or control group), double-positive cells were the population with the strongest IFN-γ production (30 and 40%, respectively). However, when cells were stimulated with the rAb ZH9F7-Cap, only the double-positive cells of the vaccinated group showed a greater IFN-γ response (approximately 4%), which was statistically significant compared to that of the rest of the groups (Figure 7d). Finally, when cells were stimulated with rAb ZH9F7 without PCV2 Cap antigen (Figure 7e), no significant differences in IFN-γ-secreting CD4^+^CD8^−^, CD4^−^CD8^+^ and CD4^+^CD8^+^ cells were found between the rAb ZH9F7-Cap and the control group (*p* > 0.05). Thus, the in vitro stimulation was triggered most likely due to the Cap antigen in rAb ZH9F7-Cap and not to the recombinant chimeric rAb ZH9F7 antibody.

## 4. Discussion

Dendritic cells play an essential role since they are considered a bridge between innate and adaptive responses. In particular, these cells can cross-present and activate naïve CD4^+^ and CD8^+^ T lymphocytes, positioning them as targets for vaccine development. Therefore, these cells have been widely studied in antigen targeting, delivering diverse antigens to DEC205^+^ cells by different routes in several species such as humans, mice, chickens, calves, and pigs [2,12,22,23,24]. Although the strategy is functional, it is unclear how different DEC205^+^ DC subpopulations are targeted and how a working mechanism could be determined for DEC205^+^ DC targeting in the swine model. Therefore, we aimed to explore how different DC populations are targeted in the blood and skin of pigs and how to relate that information to the induced humoral and cellular response after targeting the PCV2 Cap antigen.

In humans and mice, it has been observed that the engagement of an anti-DEC205 antibody with its DEC205 receptor leads to a receptor-mediated endocytosis process [11,25]. Our results confirmed that rAb ZH9F7 could trigger a receptor-mediated endocytosis process by different populations of DEC205^+^ cells from porcine blood, encompassing monocytes and the subpopulations cDC1 and cDC2. The previous finding was evidenced by a decrease in the MFI after incubation for 1 h at 37 °C, mostly in monocytes and cDC1, followed by cDC2. However, no significant differences were found among them (*p* > 0.05). This receptor-mediated endocytosis appears to be greater than that promoted by Siglec-1 on porcine macrophages [4] and similar to that promoted by the engagement of anti-CLEC12A and CLEC1A receptors on porcine cDC1 and cDC2 [26]. Auray et al., (2016) reported that all DCs in porcine blood express DEC205, although cDC1 showed the strongest expression of the receptor, followed by monocytes, cDC2, and pDC [27]. Our results agree with those reports, where cDC1 and monocytes showed higher expression of DEC205 than cDC2.

Previously, we evaluated the rAb ZH9F7 half-life in the bloodstream. The obtained results showed that after 8 h of intravenous injection, rAb ZH9F7 in sera significantly decreased [17]. This finding may be due to receptor-mediated endocytosis, a mechanism that we confirmed; therefore, rAb ZH9F7 was not found to be free in circulation but was possibly internalized in DEC205^+^ blood cells or other DEC205^+^ cells in tissues. We know that rAb ZH9F7 recognizes DEC205^+^ cDC1, cDC2 and monocytes, promoting the internalization of the rAb ZH9F7/DEC205 complex into the intracellular space more efficiently than other members of the C-type lectin family. Importantly, this result demonstrates for the first time that targeting the DEC205 receptor in a swine model induces receptor-mediated endocytosis.

Although the exact mechanisms for DC cross-presentation remain unclear, the most important step is the escape of extracellular antigens loaded in endocytic vesicles toward the cytoplasmic compartment [28]. In humans, targeting antigens coupled to anti-DEC205 antibodies leads to a process in which, after receptor-mediated endocytosis, antigens can follow a classic MHC-II pathway and escape from early endosomes, allowing intracytoplasmic localization, with access to MHC-I processing machinery [11]. In our model, intracellular localization assays after receptor-mediated endocytosis are needed to confirm whether the targeted antigens could escape from early endosomes toward the cytoplasm.

Porcine skin has been previously described as a DC-rich tissue in which different DC subpopulations with migratory abilities have been characterized [13,14]. Therefore, we aimed to explore the ability of rAb ZH9F7-Cap to recognize and target skin DCs in situ. The analysis of skin migratory DCs demonstrated a higher proportion of skin cDC2/LC migratory cells than the percentage of cDC1, which agrees with the findings reported by Marquet et al. in the evaluation of skin migratory DC subtypes [13]. Despite identifying a higher frequency of cDC2/LC in migratory cells, we found that the cDC1 population presented a higher percentage of positivity for rAb ZH9F7-Cap labeling in situ. Nonetheless, we believe that this result is justified since the expression level of DEC205 on cDCs appears to be tissue-dependent. In porcine tonsils, there was a higher expression of DEC205 on cDC1 cells than on cDC2 cells. In contrast, in the spleen, submaxillary, and mesenteric lymph nodes, cDC2 presented the highest expression of DEC205, although there were no significant differences [29]. Thus, differences in DEC205 expression levels in skin cDC1 and cDC2 could be correlated with a higher probability of in situ targeting.

It has been previously reported that targeting of DCs with poXCL1 (intradermally applied) recognized approximately 35% of skin cDC1, which was enough to induce the production of anti-M2e antibodies against porcine influenza virus [30]. The actual percentage of rAb ZH9F7-Cap-targeted DCs could be underestimated. Delozy et al. noted that it is likely that after internalization, quenching or degradation of the detection signal occurs [30]. We found a higher presence of targeted DEC205^+^ cDC2/LC in the rAb ZH9F7-Cap group than in the PBS control group. There was no difference between the targeted rAb ZH9F7-Cap group and the isotype rAb 7DEU-Cap control group. We hypothesize that this may be related to the higher capacity of cDC2 to uptake soluble antigens, as previously reported in swine blood and skin cDC2 [9,13]. Moreover, confocal microscopy experiments confirmed that the anti-DEC205 antibody binds to DEC205^+^ cells and is internalized and colocalized with other markers, such as CADM1 and CD172, which will allow us to assume the recognition by cDC1 and cDC2 from swine skin.

DCs are found in peripheral tissues, capturing and processing antigens and then migrating to lymph nodes where antigen presentation to naïve T lymphocytes and B lymphocyte activation occurs [31,32]. After in situ labeling with rAb ZH9F7-Cap and stimulation with poly (I:C), skin DCs were detected in the regional superficial inguinal lymph nodes closest to the application site. This phenomenon occurs since the lymphatic system circulation first reaches the closest regional node. Bonifaz et al., (2004) demonstrated that in mice, mAb anti-DEC205 promoted systemic targeting to reach distal lymph nodes and the spleen in a time-dependent manner. Their results showed the ability of the anti-DEC205 mAb to recognize and label DEC205^+^ DCs in several secondary lymphoid organs, which was detectable up to three days after subcutaneous injection [23]. Contrary to the findings of Bonifaz et al., we did not achieve systemic targeting. There were also significant differences in their study: the amount of antibody applied was higher, the application route was subcutaneous, and their study model was a mouse. Most likely, the subcutaneous method allows better access to the circulatory system and rapid systemic diffusion. In contrast, intradermal application results in slow and localized absorption; therefore, we used local targeting. Here, we confirmed that rAb ZH9F7 recognizes different DEC205^+^ DC populations, triggers receptor-mediated endocytosis, targets skin cDC1 and cDC2/LC, and reaches regional inguinal nodes. These findings support the notion that rAb ZH9F7 targeted cells in regional lymph nodes could be a combined effect of skin DC migration or labeling lymph node DCs in situ. In other animal models, such as mice, migration could be blocked through anti CCL19 and CCL21 antibodies or CCR7 knock-out mice. However, we did not have enough tools to evaluate this blocking effect on swine and demonstrate that migration is only due to rAb ZH9F7 targeted cells.

As a preliminary analysis, we evaluated IFN-γ-secreting cells and humoral response after intradermic application of rAb ZH9F7-Cap. We first identified different T lymphocyte populations, discerning single-positive CD4^+^ CD8^−^ and CD4^−^ CD8^+^ T cells and double-positive CD4^+^ CD8^+^ T cells. It has been previously reported that in the swine model, double-positive T cells in peripheral blood represent a larger fraction of total T lymphocytes in a mature state, reaching approximately 60% [33]. When the different T lymphocyte-identified populations were stimulated with PHA, CD4^+^ CD8^+^ cells had the highest proportion of IFN-γ-secreting cells in the vaccinated and control groups. Another interesting finding was that CD4^+^ CD8^+^ T cells from the rAb ZH9F7-Cap vaccinated group showed a higher frequency of IFN-γ secreting cells (approximately 4%) when stimulated with rAb ZH9F7-Cap. Vaccination with rAb ZH9F7-Cap promotes the differentiation of CD4^+^ CD8^+^ IFN-γ-secreting cells since these populations have strong antiviral activity against other populations of viral infections pigs [34].

In addition, detectable levels of anti-PCV2a Cap antibody higher titers were found four weeks after prime (one week after boost) in the targeted group compared to the control group and the baseline values of both groups, albeit the response was not as robust as desired. Perhaps a higher dosage or subsequent application of rAb ZH9F7-Cap could enhance the level of generated antibodies. Further studies evaluating different immunization schemes with this, or other antigens are in progress. A limitation of this study was that we did not analyze the presence of neutralizing antibodies. It may raise concerns about the protection of rAb ZH9F7-Cap against PCV2 infection. However, further studies must answer this concern.

The main advantage of using chimeric-hybrid antibodies for which the recombinant chimeric mouse × pig antibody rAb ZH9F7 was designed and expressed is in reducing the antigenicity of murine monoclonal antibodies applied in other species such as humans or pigs [17]. Therefore, when we used rAb ZH9F7 without Cap antigen for in vitro stimulation experiments, we confirmed that the induction of IFN-γ-secreting cells was not significant in response to rAbZH9F7 itself. Thus, it is remarkable that hybrid antibodies with prophylactic or therapeutic uses potentially reduces the immune response.

Of note, skin cDC1 showed a more significant response to targeting with rAb ZH9F7-Cap in situ, possibly skewing the response to a Th1 profile. The abovementioned finding supports the generation of CD4^+^ CD8^+^ IFN-γ^+^ cells, representing activated and memory Th1 CD4^+^ T cells that express the CD8α receptor [35]. However, a limitation of our study is that only the Th1 response was evaluated by measuring intracellular IFN-γ. Considering other cytokines, such as IL-10, IL-8, IL-12, TNF-α, TGF-β, and IL1-β, would help further understand targeting to cDC1 and whether cDC2/LC and moDC/Macro cause a skew in the profile of the immune response.

## 5. Conclusions

In this study, we confirmed the functionality of the antigen targeting of rAb ZH9F7 anti-porcine DEC205 and its effect on different populations of blood and skin porcine DEC205^+^ DCs. rAb ZH9F7 promotes efficient receptor-mediated endocytosis in blood cDC1, cDC2, and monocytes and can better recognize skin cDC1 than cDC2/LC and moDCs/Macro. Moreover, these DC populations could reach the local lymph nodes across the lymph drainage system and prime naïve T cells, inducing the response of double-positive CD4^+^CD8^+^ IFN-γ secreting cells, which have been recognized as porcine memory T cells with an essential role in the antiviral response. These results undoubtedly help outline how the antigen-targeting DC DEC205^+^ system works in a porcine model.

## Figures and Tables

**Figure 1 vaccines-10-00684-f001:**
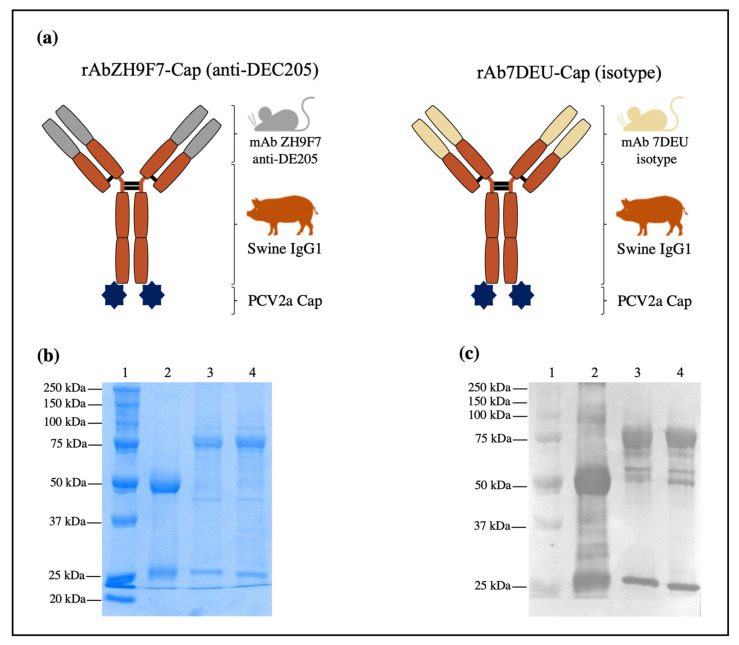
Design, expression, and characterization of rAb ZH9F7-Cap and isotype control 7DEU-Cap. (**a**) Schematic illustration of the recombinant mouse × pig chimeric rAb ZH9F7-Cap and isotype control rAb 7DEU-Cap. (**b**) 10% SDS–PAGE followed by Coomassie blue staining of purified rAb ZH9F7-Cap and rAb 7DEU-Cap. Lane 1: molecular weight marker; lane 2: rAb ZH9F7 without Cap; lane 3: purified rAb ZH9F7-Cap; lane 4: purified rAb 7DEU-Cap. (**c**) Western blot of the chimeric mouse × pig rAb ZH9F7-Cap and rAb 7DEU-Cap. Lane 1: molecular weight marker; lane 2: rAb ZH9F7 without Cap; lane 3: purified rAb ZH9F7-Cap; lane 4: purified rAb 7DEU-Cap. HRP-conjugated goat anti-porcine IgG (H + L) was used to confirm the expression of both mouse × pig recombinant antibodies.

**Figure 2 vaccines-10-00684-f002:**
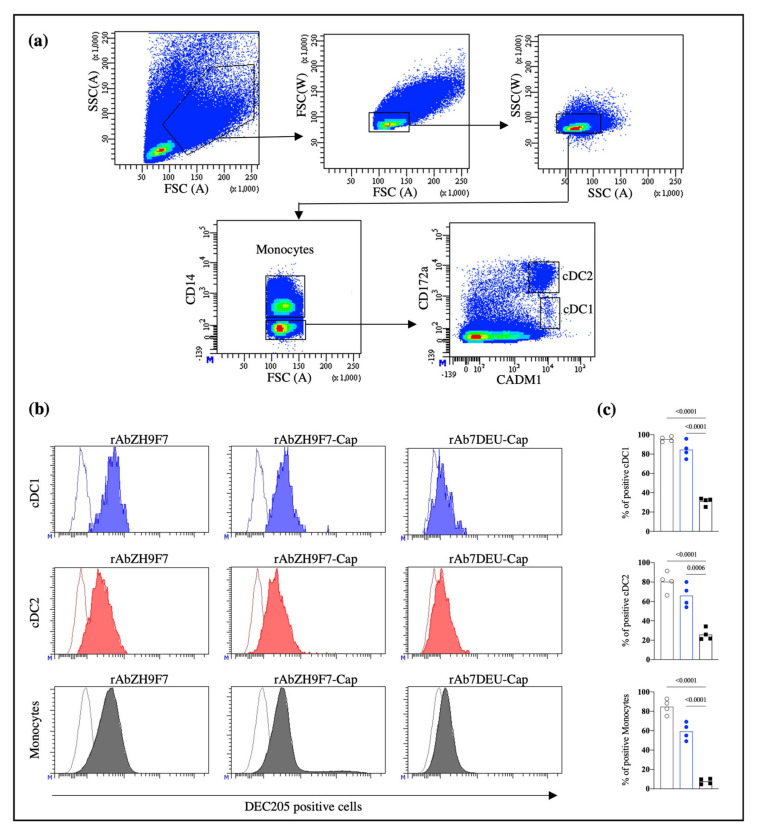
Evaluation of the activity of rAb ZH9F7-Cap to recognize DEC205^+^ mononuclear cells in peripheral blood (PBMCs) from pigs. (**a**) Strategy for identifying monocytes, cDC1, and cDC2 in blood circulation. (**b**) Identification of DEC205^+^ cDC1, cDC2, and monocytes after staining with rAb ZH9F7, rAb ZH9F7-Cap, and isotype control rAb 7DEU-Cap. Contour histograms illustrate the FMO, and colored histograms illustrate the expression of DEC205 on cDC1 (blue), cDC2 (red), and monocytes (gray). Histograms represent the results of one of four animals. (**c**) Graphic analysis of the percentages of positive cDC1 (top), cDC2 (middle), and monocyte (bottom) cells. rAb ZH9F7 (gray); rAb ZH9F7-Cap (blue); Isotype control rAb 7DEU-Cap (black). One-way ANOVA and the Tukey–Kramer test for multiple comparisons of means were performed (*p* < 0.05). *p*-values denote statistically significant differences in the means of the different groups.

**Figure 3 vaccines-10-00684-f003:**
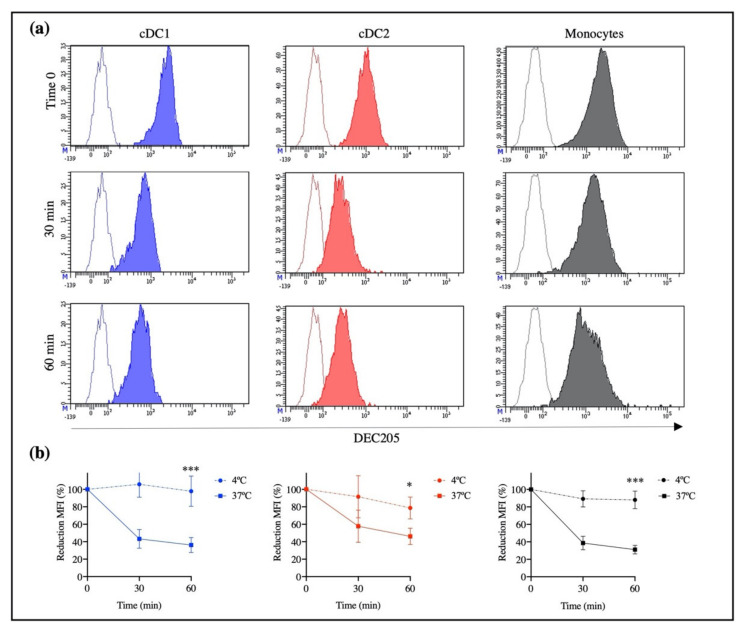
Identification of cDC1, cDC2, and DEC205^+^ monocytes by rAb ZH9F7-Cap and receptor-mediated internalization. DEC205 surface expression in cDC1, cDC2, and monocytes after 30 and 60 min of incubation with rAb ZH9F7 at 37 °C. (**a**) Contour histograms represent FMO, while colored histograms represent the expression of DEC205 on cDC1 (blue), cDC2 (red), and monocytes (gray). (**b**) Percentage of mean fluorescence intensity (MFI) after incubation at 37 °C and 4 °C by cDC1 (blue), cDC2 (red), and monocytes (black). Dots represent the mean ± SD of reduction in MFI on cells obtained from 3 pigs. Asterisks denote statistically significant differences between MFI after incubation at 37 °C and 4 °C for 1 h (*** *p* < 0.0005; * *p* < 0.05).

**Figure 4 vaccines-10-00684-f004:**
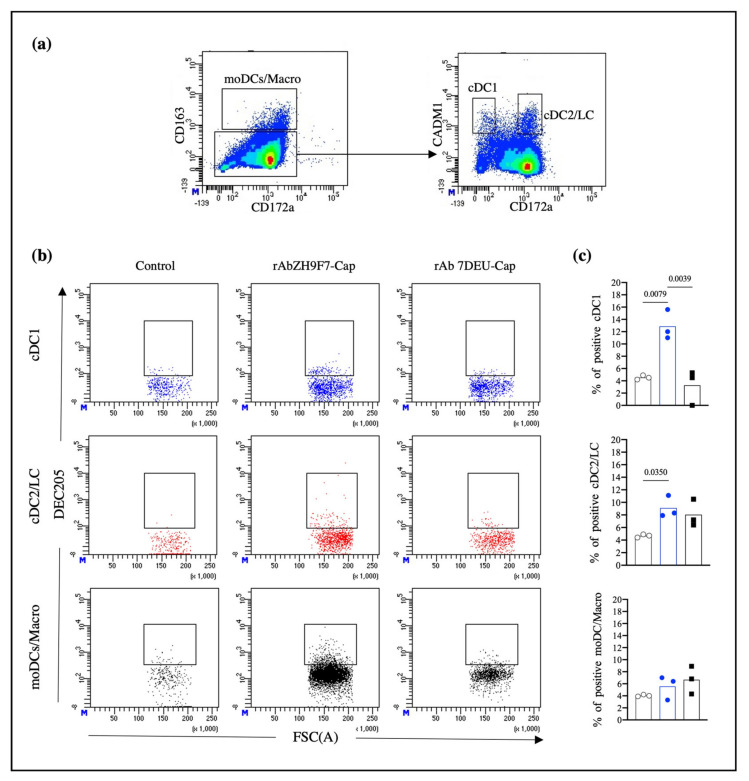
rAb ZH9F7-Cap targets skin cDC1 s and cDC2s/LCs in vivo. (**a**) Skin migratory CD163^−/low^CD172a^+/−^ cells were selected and analyzed for CADM1 expression, obtaining skin cDC1 (CD163^−^CADM1^+^CD172a^−/low^) and cDC2/LC (CD163^−/low^CADM1^+^CD172a^high^) cells. (**b**) Skin cDC1, cDC2/LC, and moDCs/Macro positive for targeting in vivo. The dot plot represents results from one of three experimental units from the control, rAb ZH9F7-Cap, and rAb 7DEU-Cap. (**c**) Average percentages of cDC1, cDC2/LC, and moDCs/Macro positive from in vivo targeting from three groups. PBS control (gray); rAb ZH9F7-Cap (blue); Isotype control rAb 7DEU-Cap (black). One-way ANOVA and the Tukey–Kramer test for multiple comparisons of means were performed (*p* < 0.05). *p*-values denote statistically significant differences in the means of the different groups.

**Figure 5 vaccines-10-00684-f005:**
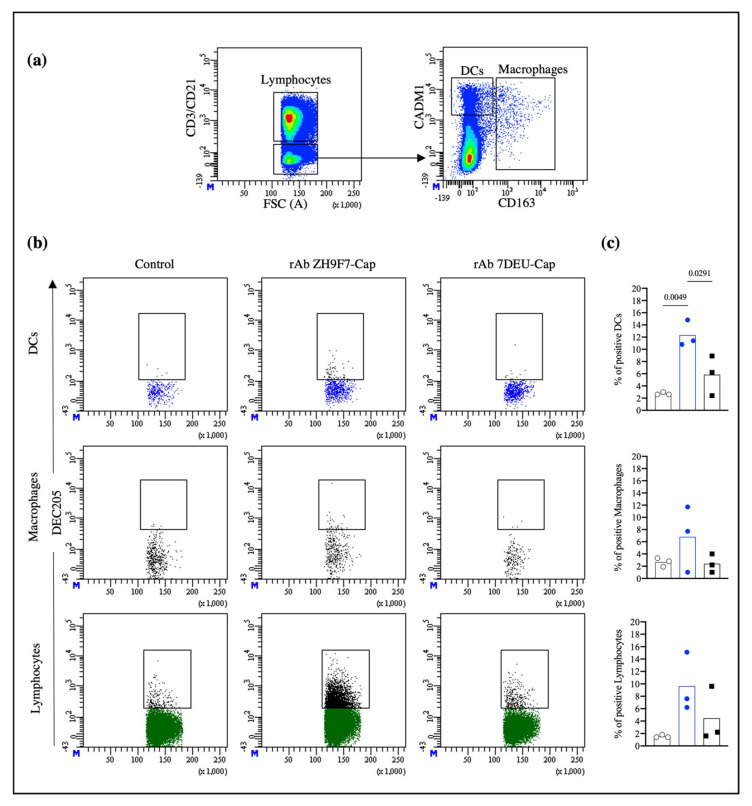
Targeted skin DCs and lymphocytes migrate to regional inguinal nodes. (**a**) Strategy to analyze DEC205^+^-targeted cells in superficial inguinal lymph nodes. The CD3^−^ CD21^−^ population was analyzed for CD163 and CADM1 expression (bottom panel) and to identify potential DCs (CD3^−^ CD21^−^CADM1^+^ CD163^−^). (**b**) Analysis of rAb targeted DEC205^+^ cells from the control, rAb ZH9F7-Cap, and rAb 7DEU-Cap. (**c**) Average percentages of DCs and macrophages. PBS control (gray); rAb ZH9F7-Cap (blue); Isotype control rAb 7DEU-Cap (black). One-way ANOVA and the Tukey–Kramer test for multiple comparisons of means were performed (*p* < 0.05). *p*-values denote statistically significant differences in the means of the different groups.

**Figure 6 vaccines-10-00684-f006:**
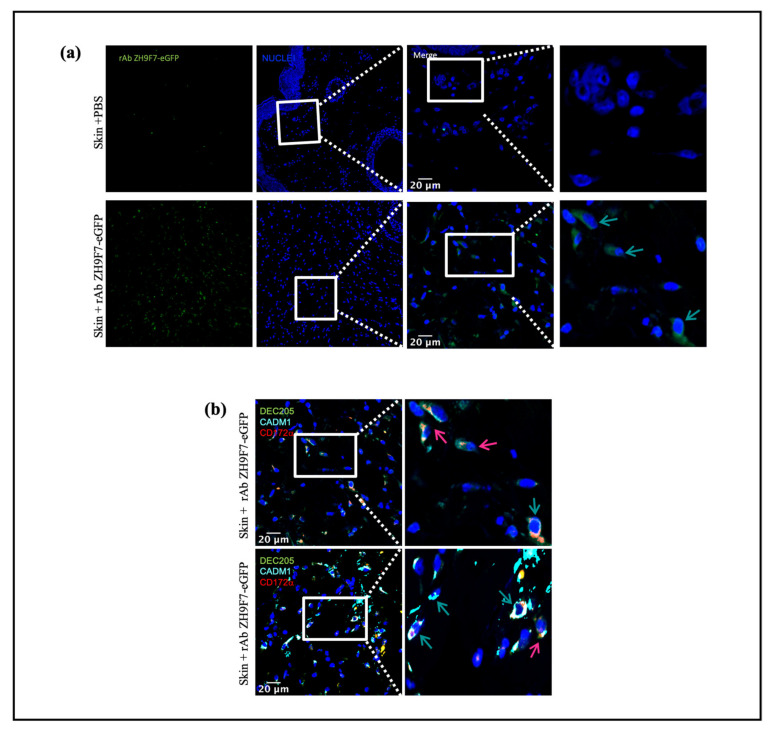
Skin DEC205 targeted DCs. Representative immunofluorescence micrographs from targeted DEC205 DCs were evaluated. (**a**) Control and rAb ZH9F7-eGFP treated skin DEC205^+^ cells (green) are shown in left panels, and zoomed areas (scale bar 20 µm) are shown in right panels. Identification of targeted DEC205 DC subsets by expression of (**b**) DEC205^+^CADM1^+^, cDCs1 (pink arrows) and DEC205^+^CADM1^+^CD172a^+^, cDCs 2 (bluish-green arrow). Representative immunofluorescence micrographs from a zoomed area (scale bar = 20 µm). Data from different regions of biopsies.

**Figure 7 vaccines-10-00684-f007:**
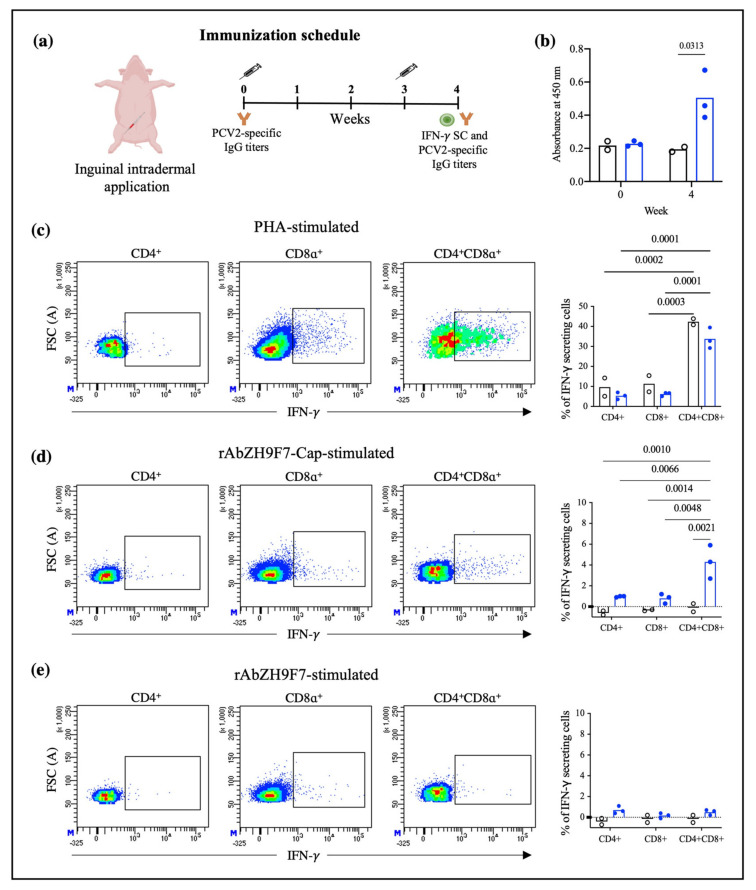
Immune response promoted by intradermal application of rAb ZH9F7-Cap in swine. (**a**) Immunization schedule. (**b**) Indirect ELISA for the detection of PCV2a anti-Cap antibodies. Serum samples were tested for anti-PCV2a Cap IgG detection at weeks 0 and 4 postimmunization. (**c**) Dot plots represent the intracellular IFN-γ in cells stimulated with PHA, rAb ZH9F7-Cap (**d**), and rAb ZH9F7 without PCV2 Cap as a control (**e**). In (**b**–**d**), graphs represent the average percentage of IFN-γ expression by the CD4^+^CD8^−^, CD4^−^CD8^+^, and CD4^+^CD8^+^ populations. Black bars-dots represent cells from the control group (*n* = 2), and blue bars-dots represent cells from the rAb ZH9F7-Cap vaccinated group (*n* = 3). Two-way ANOVA and Tukey–Kramer tests for multiple comparisons of means were performed (*p* < 0.05). *p*-values denote statistically significant differences in the means of the different treatments. In (**b**–**d**), the basal expression of IFN-γ from unstimulated cells was subtracted.

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
