# Peer review of "Antigen Targeting of Porcine Skin DEC205+ Dendritic Cells"

_vaccines, 2022, doi:10.3390/vaccines10050684_

Round 1
Reviewer 1 Report
Dendritic cells (DCs) play an important role as they are considered a bridge between the innate and adaptive responses. DCs targeting via DEC205+ cells had been shown to be effective in promoting the internalization of antigens that may trigger a specific immune response. Previously, the research group developed a recombinant mouse x pig chimeric antibody against porcine DEC205 receptor (rAb ZH9F7), and antigens from PRRSV were targeted by intradermal application. Their results showed that the system is immunogenic with limited protective immunity. In this study, they characterized antigen targeting in a swine model and evaluated the ability of the rAb ZH9F7 to be internalized by DEC205+ DCs in peripheral blood, applied the rAb ZH9F7 recombinant antibody intradermally and evaluated targeting of dermal cDC1 and cDC2 as well as the presence of targeted DCs in regional lymph nodes, and the antigenized rAb ZH9F7_Cap promoted antibody production against PCV2a Cap and the differentiation of double-positive CD4+CD8+ cells that respond to stimulation with rAb ZH9F7_Cap through IFN production characterized by its strong antiviral capabilities. Their finding has guiding significance for developing vaccines based on antigen targeting to DCs.
However, the manuscript needs significant grammatical editing as the current state of the text makes it difficult to follow.
Author Response
Thank you for the positive comments on our work.
The manuscript needs significant grammatical editing as the current state of the text makes it difficult to follow.
R: The changes were done accordingly.
Reviewer 2 Report
Melgoza-González and colleagues investigated the ability of a the recombinant antibody rAbZH9F7 (anti-DEC205) to either trigger DC endocytosis (in vitro experiments) and to promote the development of a Th1 response (in vivo experiments). First, in vitro experiments were carried out using PBMC and researchers observed that the tested recombinant antibody specifically recognized DCs and triggered a receptor-mediated endocytosis process in cDC1, cDC2 and monocytes from porcine blood. In vivo studies were then carried out, through intradermal application of the antigenized rAbZH9F7 combined with PCV2 cap antigen (rAbZH9F7_Cap). Researchers observed that rAbZH9F7 recognized DEC205+ cells (cDC, monocytes), promoted receptor-mediated endocytosis and migration of cDCs to the regional lymph nodes, with subsequent development of a Th1 response (evidenced by the proliferation of CD4+CD8+ T cells).
The work is well-structured and address an interesting topic. Results generated in this study are relevant for the scientific community.
Nevertheless, there are diverse points that should be addressed before considering this paper suitable for publication:
Abstract
- Line 32: ‘CD4+CD8+ cells’ --> ‘CD4+CD8+ T cells’ ?
Introduction
- Line 48-50: It should be useful to the reader to understand why and how moDC are generated (e.g. they are generated from blood monocytes due to the low frequency of DC in blood and tissues)
Materials and methods
- Line 97: number of pigs used in the study (for both in vitro and in vivo studies)
- Brands of some reagents are missing.
- PBMC purification and culture should be slightly better described in a separate subparagraph of the material and method section.
- 136-161: Please state which flow cytometer did you use, number of events acquired, software for analysis, gating strategy….
- Line 190: describe the ‘antibiotic-antimycotic cocktail’
- Line 190-191: please briefly describe how the enrichment of DCs was performed by an OptiPrep density gradient.
- Line 228: how many mL did you collect? EDTA blood? Did you use the plasma for ELISA described at 230-245 or did you collect blood in other tube for sera?
- Line 262: 30 min? RT or 4°C?
- Line 262-271: did you check normality with graph pad as well? Number of replicates for in vitro and in vivo studies?
Figure 4 and 5: Number of events acquired.
Figure 7, panel c-d-e: I apologise but do not completely understand this figure. Did you stimulate the PBMC of a treated pig 4 week post-injection? There was a control?
Supplementary figure 5: Why did you not use CD3 in your gating strategy?
Discussion: Maybe just briefly add some possible implication of your discovery, such as use of recombinant antibody rAbZH9F7 in vivo?
Author Response
Thank you for the positive comments on our work.
Abstract
Line 32: ‘CD4+CD8+ cells’ --> ‘CD4+CD8+ T cells’?
Answer: Yes, we referred to double-positive T cells. The changes were done accordingly.
Introduction
Line 48-50: It should be useful to the reader to understand why and how moDC are generated (e.g. they are generated from blood monocytes due to the low frequency of DC in blood and tissues)
Answer: For the mentioned experiment, in vitro-derived moDCs were generated from blood monocytes cultured in the presence of interleukin-4 and GM-CSF.
Materials and methods
Line 97: number of pigs used in the study (for both in vitro and in vivo studies)
Answer: Nineteen pigs were used in this study:
- 3 for internalization studies by flow cytometry
- 11 for skin and lymph nodes targeting
- 5 to evaluate immunization
Brands of some reagents are missing.
Answer:: The changes were done accordingly.
PBMC purification and culture should be slightly better described in a separate subparagraph of the material and method section.
Answer: Briefly, 4 mL of EDTA blood samples were added carefully into a 15 ml tube with 8 mL of Ficoll reagent, using a ratio blood-Ficoll Paque 1:2. Then, the tubes were centrifuged at 1600 rpm, for 30 min without a break. After centrifugation, the PBMCs interface was carefully taken and washed twice with RPMI-1640 supplemented with an antibiotic-antimycotic cocktail, 10% FBS, and 2 mM EDTA.
136-161: Please state which flow cytometer did you use, number of events acquired, software for analysis, gating strategy....
Answer: Missing information about cytometry, software, and the number of events acquired was included accordingly.
Line 190: describe the ‘antibiotic-antimycotic cocktail’
Answer: The antibiotic-antimycotic cocktail was described, indicating each reagent's final concentration.
Line 190-191: please briefly describe how the enrichment of DCs was performed by an OptiPrep density gradient.
Answer:: A more detailed description of Opti Prep gradient enrichment was done accordingly.
Line 228: how many mL did you collect? EDTA blood? Did you use the plasma for ELISA described at 230-245 or did you collect blood in other tube for sera?
Answer: we collected 6 mL of peripheral blood on tubes with EDTA to evaluate cellular response and other 6 mL of blood without anticoagulant to antibody response. The changes were done accordingly.
Line 262: 30 min? RT or 4°C?
Answer: All 30 min incubations were performed at room temperature.
Line 262-271: did you check normality with graph pad as well?
Answer: Yes, normality was determined and confirmed using the Shapiro-Wilk test on the GraphPad prism.
Number of replicates for in vitro and in vivo studies?
Answer: In both in vitro and in vivo studies, at least three biological replicates were performed in almost all assays unless indicated in each section.
Figure 4 and 5: Number of events acquired.
Answer: For the analysis presented in figure 5, near 500x103 events were acquired. On the other hand, analysis in figure 4 was carried outreaching at least 130x103 until around 470x103 events. It is worth mentioning that the yield in the skin migratory cells harvesting was low. Thus, only 500 x103 cells were stained per tube.
Figure 7, panel c-d-e: I apologize but do not completely understand this figure. Did you stimulate the PBMC of a treated pig 4 week post-injection?
Answer: Yes, four weeks after immunization (one week after boost), we take blood samples to isolate, culture, and stimulate PBMCs from the rAbZH9F7_Cap vaccinated group and control group (n=3 and n=2, respectively).
There was a control?
Answer: We have a control group without vaccination with rAb ZH9F9_Cap, and another vaccinated with rAbZH9F7_Cap. In the conditions for in vitro stimulation, we have the PHA positive control to IFN-g expression (7c), rAb ZH9F7_Cap-stimulated cells (7d), and rAb ZH9F7 without Cap antigen-stimulated cells (7e). The graphs represent the percentages of IFN-g positive cells where basal expression of IFN-g from non-stimulated cells was subtracted.
Supplementary figure 5: Why did you not use CD3 in your gating strategy?
Answer:: We were unable to include CD3 labeling due to the availability of our secondary antibodies. Although the expression of CD3 could help us to discriminate between NK and T lymphocytes, our results showed that only the double-positive CD4+CD8+ T cells were stimulated to produce IFN-g whereas porcine NK cells have a CD4- phenotype. Thus, despite the absence of CD3 on our strategy gating, this did not impair the obtained results and conclusions.
Discussion: Maybe just briefly add some possible implication of your discovery, such as use of recombinant antibody rAbZH9F7 in vivo?
Answer: The changes were done accordingly.
Reviewer 3 Report
Comments on vaccines-1660741
(Title: Antigen targeting to porcine skin DEC205+ dendritic cells)
In the present study, Edgar Alonso Melgoza-González et al. generated a fusion antigen harboring a recombinant mouse x pig chimeric antibody (rAbZH9F7) against the C-type lectin family member receptor DEC205 and the Cap protein of porcine circovirus type 2 (PCV2) (rAbZH9F7_Cap) and evaluated the immune response in pigs. They showed that rAbZH9F7 was able to recognize DEC205+ cells, including conventional dendritic cells (DCs), and monocyte-derived DCs (moDCs) from the blood and skin but also promoted receptor-mediated endocytosis and migration of cDCs and moDCs toward regional lymph nodes and intradermal vaccination of pigs with rAbZH9F7_Cap induced a higher frequency of IFN-secreting CD4+CD8+ T lymphocytes and anti-Cap antibodies than those of the control group. Generally, the study was well designed and presented and the work is interesting and potentially useful. Several concerns should be addressed.
- The receptor recognition and cellular endocytosis and migration of the rAbZH9F7-recongzied DEC205+ DCs should be demonstrated by additional assays.
- The immune responses induced by the vaccination need to be evaluated by well-established methods including ELISA/ELIspot kits.
- Serum PCV2-specific neutralizing antibodies should be examined by virus neutralization test.
- The manuscript should be improved by native English speakers. Try to avoid inconsistency (eg. rAbZH9F7 vs rAb ZH9F7; rAbZH9F7_CaprAb ZH9F7_Cap, etc.).
Author Response
Thank you for the positive comments on our work.
- The receptor recognition and cellular endocytosis and migration of the rAbZH9F7-recognized DEC205 DCs should be demonstrated by additional assays.
Answer: Although we have performed an internalization assay similar to that previously reported by (Álvarez et al., 2020), we agree with the limitation of indirect receptor-mediated endocytosis evaluation through flow cytometry on blood cDC1, cDC2, and monocytes subsets. Nevertheless, we assume that images by confocal microscopy support and confirm the internalization of rAb ZH9F7 by swine skin DCs. Regarding experiments to confirm the migration of in situ-targeted DCs, we don’t have a strategy to demonstrate that this effect was only due to in situ-targeting. It has previously described the possibility that the anti-DEC205 antibody could diffuse toward lymph afferent or gain access to the bloodstream, reaching distal lymph nodes (Bonifaz et al., 2004). Thus, targeted cells in lymph nodes could be a synergic effect of DCs migrating from the skin and targeting DEC205+ cells in regional lymph nodes.
- The immune responses induced by the vaccination need to be evaluated by well-established methods including ELISA/ELIspot kits.
Answer: The main advantage in evaluating cellular response through flow cytometry is that we cannot only determine the frequency of IFN-secreting cells but also identify the phenotype of T cell stimulated. Furthermore this strategy has been used in other vaccine evaluations (Ferrari et al., 2014; Koinig et al., 2015), our research group had successfully used it (Bustamante-Córdova et al., 2019). On the other hand, we developed the in-house indirect ELISA using the identical Cap antigen, which was targeted through the rAb ZH9F7 anti-DEC205.
- Serum PCV2-specific neutralizing antibodies should be examined by virus neutralization test.
Answer: One limitation of the present study naturally includes the absence of neutralizing antibodies quantification since we were not able to carry out this determination.
- The manuscript should be improved by native English speakers. Try to avoid inconsistency (eg. rAbZH9F7 vs rAb ZH9F7; rAbZH9F7_CaprAb ZH9F7_Cap, etc.).
Answer: The changes were done accordingly.
References
Bonifaz, L. C., Bonnyay, D. P., Charalambous, A., Darguste, D. I., Fujii, S.-I., Soares, H., . . . Steinman, R. M. (2004). In vivo targeting of antigens to maturing dendritic cells via the DEC-205 receptor improves T cell vaccination. The Journal of experimental medicine, 199(6), 815-824.
Bustamante-Córdova, L., Reséndiz-Sandoval, M., & Hernández, J. (2019). Evaluation of a Recombinant Mouse X Pig Chimeric Anti-Porcine DEC205 Antibody Fused with Structural and Nonstructural Peptides of PRRS Virus. Vaccines, 7(2), 43.
Ferrari, L., Borghetti, P., De Angelis, E., & Martelli, P. (2014). Memory T cell proliferative responses and IFN-γ productivity sustain long-lasting efficacy of a Cap-based PCV2 vaccine upon PCV2 natural infection and associated disease. Veterinary research, 45(1), 1-16.
Koinig, H. C., Talker, S. C., Stadler, M., Ladinig, A., Graage, R., Ritzmann, M., . . . Saalmüller, A. (2015). PCV2 vaccination induces IFN-γ/TNF-α co-producing T cells with a potential role in protection. Veterinary research, 46(1), 1-13.
Álvarez, B., Nieto-Pelegrín, E., Martínez de la Riva, P., Toki, D., Poderoso, T., Revilla, C., . . . Domínguez, J. (2020). Characterization of the porcine CLEC12A and analysis of its expression on blood dendritic cell subsets. Frontiers in immunology, 11, 863.
Round 2
Reviewer 3 Report
Sorry, I don't think the major concerns have been addressed adequately.
Author Response
In the present study, Edgar Alonso Melgoza-González et al. generated a fusion antigen harboring a recombinant mouse x pig chimeric antibody (rAbZH9F7) against the C-type lectin family member receptor DEC205 and the Cap protein of porcine circovirus type 2 (PCV2) (rAbZH9F7_Cap) and evaluated the immune response in pigs. They showed that rAbZH9F7 was able to recognize DEC205+ cells, including conventional dendritic cells (DCs), and monocyte-derived DCs (moDCs) from the blood and skin but also promoted receptor-mediated endocytosis and migration of cDCs and moDCs toward regional lymph nodes and intradermal vaccination of pigs with rAbZH9F7_Cap induced a higher frequency of IFN-secreting CD4+CD8+ T lymphocytes and anti-Cap antibodies than those of the control group. Generally, the study was well designed and presented and the work is interesting and potentially useful. Several concerns should be addressed.
- The receptor recognition and cellular endocytosis and migration of the rAbZH9F7-recongzied DEC205+ DCs should be demonstrated by additional assays.
- The immune responses induced by the vaccination need to be evaluated by well-established methods including ELISA/ELIspot kits.
- Serum PCV2-specific neutralizing antibodies should be examined by virus neutralization test.
- The manuscript should be improved by native English speakers. Try to avoid inconsistency (eg. rAbZH9F7 vs rAb ZH9F7; rAbZH9F7_CaprAb ZH9F7_Cap, etc.).
First, we would like to thank the Reviewer for taking the time to review and criticize our study. We appreciate that the Reviewer remarked that our research was well designed and presented and that, in the opinion of Reviewer 3, our work is interesting and potentially useful.
Response to the four Reviewer's concerns:
1. The receptor recognition and cellular endocytosis and migration of the rAbZH9F7-recongzied DEC205+ DCs should be demonstrated by additional assays.
A: We partially agree with the Reviewer’s comment. We use flow cytometry and confocal fluorescence microscopy to probe the recognition and endocytosis. We examined it on PBMC and directly on the skin. Indeed, we partially demonstrated the migration of targeted-DCs. Still, the tools available to evaluate the immune response in pigs are limited, and it was impossible to perform additional experiments, such as using antibodies against chemokine receptors. A strategy to block DCs migration is using anti-CC21 and anti CCL19 antibodies, which avoid the response of chemokine receptor CCR7 on the DC surface, thus inhibiting their migratory response. Unfortunately, these experiments are expensive and difficult to perform. On the other hand, using an anti-porcine CCR7 or CCR7 knock-out pigs is not suitable for these assays. We will appreciate your comments if the Reviewer 3 knows other methods that we can use in the swine model.
2. The immune responses induced by the vaccination need to be evaluated by well-established methods including ELISA/ELIspot kits.
A: We disagree with the Reviewer`s comment. This experiment demonstrated which T cells are primed using this chimeric antibody, anti-porcine DEC205. ELISA or ELIspot kits will not provide this information. If the aim is to know the amount of IFN-g, independent of the source of the cells, these methods will be the first election. Once we see the kind of cells that produce IFN-g, the next step is to evaluate the amount of IFN-g that this strategy can induce.
3. Serum PCV2-specific neutralizing antibodies should be examined by virus neutralization test.
A: We partially agree with the reviewer’ comment. Neutralizing antibodies should be examined. But as previously described in the previous statement, we seek to identify the seroconversion for the moment. Further studies should evaluate neutralizing antibodies as well as their protection. In addition, neutralizing antibodies against PCV2 is not a conventional assay, and it was not possible to perform at this moment.
4. The manuscript should be improved by native English speakers. Try to avoid inconsistency (eg. rAbZH9F7 vs rAb ZH9F7; rAbZH9F7_CaprAb ZH9F7_Cap, etc.).
A: American Journal Experts have edited the manuscript (I have attached the AJE Editing Certificate). If, in the opinion of Reviewer 3, our manuscript still needs to be improved by a native English speaker, please confirm it.
Round 3
Reviewer 3 Report
My suggestions have been provided for your consideration. I have no more commmets.
Author Response
Thank you.